# Nutritional care for children with feeding difficulties and disabilities: A scoping review

**Alyssa Klein** **\*, Malia Uyehara , Andrew Cunningham, Madina Olomi, Kristen Cashin, Catherine M. Kirk**

USAID Advancing Nutrition, Arlington, Virginia, United States of America

\* alyssa_klein@jsi.com

**Data Availability Statement:** Our data has been submitted to the USAID Data Development Library, which is an open data repository for USAID-funded work. USAID will review the submission and it will

## Abstract

One billion people worldwide have a disability, and 80 percent of them live in low- and middle-income countries (LMICs). The prevalence of feeding difficulties globally ranges from 25–45 percent to 33–80 percent in children without and with disabilities, respectively. The U. S. Agency for International Development's (USAID) flagship multi-sectoral nutrition project, USAID Advancing Nutrition, conducted a scoping review of programs supporting nutritional care of children with disability and non-disability related feeding difficulties. The non-systematic scoping review included a desk review of peer-reviewed and non-peer-reviewed literature and key informant interviews. In all, 127 documents with publication dates ranging from 2003 to 2022 were identified through keyword searches and snowballing and met the inclusion criteria, and 42 experts in nutrition and disability were interviewed. Findings were organized using structured matrices of challenges and opportunities across the universal progressive model of care framework in the identification and management of feeding difficulties and disabilities and support for children with feeding difficulties and disabilities and their families. The review found insufficient policies, programs, and evidence to support children with feeding difficulties and disabilities and their families. While some resources and promising approaches exist, they are not standardized or universally used, staff are not trained to use them, and there is insufficient funding to implement them. The combination of challenges in identifying feeding difficulties and disabilities, a lack of understanding of the link between disabilities and feeding, and weak or nonexistent referral or specialized services puts these children at risk of malnutrition. Additionally, their families face challenges providing the care they need, including coping with high care demands, accessing support, obtaining appropriate foods, and managing stigma. Four areas of recommendations emerged to support children with feeding difficulties and disabilities: (1) Strengthen systems to improve identification and service provision; (2) Provide direct support to families to address determinants that affect nutrition outcomes; (3) Conduct advocacy to raise awareness of the needs and opportunities; and (4) Build the evidence base on effective interventions to identify and support these children and their families.

be posted online after it clears that process. Reference number: 435-9.

**Funding:** This scoping review was conducted for the United States Agency for International Development (USAID). It was prepared under the terms of contract 7200AA18C00070 awarded to JSI Research & Training Institute, Inc. (JSI). The contents are the responsibility of JSI and do not necessarily reflect the views of USAID or the U.S. Government. The funders were involved in study design and reviewed the paper prior to publication.

**Competing interests:** I have read the journal's policy and the authors of this manuscript have the following competing interests: CMK has been involved in the implementation and evaluation of the Multi-Agency International Training and Support (MAITS) Working with Infants with Feeding Difficulties and Baby Ubuntu programmatic approaches in Rwanda. The authors have no other conflicts of interest to report.

## Introduction

At least one in eight people worldwide—one billion people—have disabilities (WHO and World Bank 2011). Eighty percent live in low- and middle-income countries (LMICs), and nearly 100 million are children [1, 2]. "Persons with disabilities include those with long-term physical, mental, intellectual, or sensory impairments which, in interaction with various barriers, may hinder their full and effective participation in society on an equal basis with others" [3]. These barriers include factors such as negative attitudes, inaccessible transportation and infrastructure, limited social support, poor health systems, lack of access to assistive technology, inaccessible information and communications, and discriminatory legislation and policies.

Good nutrition is essential for child development. Malnutrition can cause disability in the short and long term, while disability can also lead to malnutrition. For example, maternal micronutrient deficiency increases neural tube defects (folate); impaired cognitive development (iodine, iron); as well as preterm birth and the associated increased incidence of cerebral palsy and cognitive, vision, and hearing disabilities (vitamin D, calcium) [4]. Infants and young children who are stunted, underweight, wasted, and/or suffer a micronutrient deficiency have increased incidence of physical, sensory, and cognitive disabilities [4]. Disability, in turn, can impact nutrient absorption or nutrient needs because of the way some disabilities impact the body, such as difficulties in feeding or poorer absorption of nutrients [5]. Children with disabilities may also be neglected due to stigma or misinformation, which can increase their risk of malnutrition [5]. A systematic review found that children with disabilities were three times more likely to be underweight and twice as likely to be stunted or wasted than non-disabled children [6]. Children with disabilities are more likely to be overweight or obese [6] or anemic [7]. They are twice as likely as non-disabled children to die from malnutrition [2]. Children with disabilities living in LMICs may be more at risk of malnutrition due to the failure of public health systems to provide nutritional support for children with feeding difficulties [6].

The prevalence of feeding difficulties in the global pediatric population ranges from 25–45 percent in non-disabled children and 33–80 percent in children with disabilities [8]. Additionally, complications of feeding difficulties can include aspiration and choking [9], lung infections [10], delays in developing feeding skills [11], stress on families, and extended time for meals [12]. A lack of resources, equipment, and access to services can create severe emotional and financial strain on families and communities; this strain is exacerbated for children with disabilities and their families by the stigma surrounding disability [13].

These numbers demonstrate the importance of providing adequate nutritional care and support for children with feeding difficulties and disabilities. However, there are no global reviews of systems, initiatives, and programs for improving nutritional care for these children. To address this gap, the U.S. Agency for International Development's (USAID) flagship multi-sectoral nutrition project, USAID Advancing Nutrition, conducted a non-systematic scoping review of programs and evidence focused on supporting nutritional care of children with disability-related and non-disability-related feeding difficulties. We reviewed policies, guidelines, and programs provided through health systems, and looked at existing community and family supports, as well as any gaps. Based on the review findings, we developed recommendations for opportunities to improve support for children with feeding difficulties, especially children with disabilities.

## Methods

The scoping review included a desk review of peer-reviewed and gray literature (project reports, briefs, and evaluations), as well as existing intervention tools, and key informant

interviews with experts in nutrition and disability. The scoping review method was selected over a systematic review given the need to more broadly and flexibly review the peer reviewed and non-peer reviewed literature and tools on the subject [14].

## Desk review

**Eligibility criteria.** The desk review had no date limitation. Identified resources included literature published between 2003 and 2022 and primarily literature pertaining to children ages zero–five years, though some targeted broader age ranges. The review included literature related to breastfeeding and complementary feeding practices primarily in LMICs, though relevant literature from high-income contexts was included since services could be adapted to other contexts. It specifically looked at the following conditions:

- persistent feeding difficulties in all children

- feeding difficulties and nutrition among children with disabilities

- small and sick newborns

- malnutrition treatment among children with a feeding difficulty, whether or not they had a disability.

**Information sources.** The primary document search was conducted from Dec 2020-May 2022 using a non-systematic approach for review. Search terms were based on our inclusion criteria, such as "feeding difficulties," "nutrition" or "feeding" and "disability," "small and sick newborn feeding," "cerebral palsy" and "feeding" or "nutrition." The study team used these terms to search for relevant peer reviewed and non-peer reviewed studies and other articles and programmatic resources. Titles, abstracts, and summaries of all identified resources were screened by the review team, and only relevant titles were included for a full document review. The following sources were searched by the study team using the above search terms:

- Google Scholar

- Emergency Nutrition Network database (en-net.org)

- Better Care Network resources.

We also requested resources from working groups focused on early childhood development, health, and nutrition by sending an email inviting group members to share resources directly with the study team, and all resources shared were reviewed by the study team. The working groups included—

- Early Childhood Development task force for Global Partnership on Children with Disabilities (GPcwd)

- Two CORE Group sub-groups (nutrition and disability inclusive health).

Following this initial search, an iterative process was used to search for additional topics raised in interviews that were not captured in the initial document review, for example specifically reviewing literature on cleft lip/palate in Google Scholar. In addition, documents shared by key informants were also reviewed. Of 187 potential resources identified, 127 (67.9 percent) met inclusion criteria (see Fig 1).

Resources focused primarily on LMICs and publication dates ranged from 2003–2022 with 76 percent of resources published since 2015. The majority of eligible resources were research articles or published books (n = 84, 66.1 percent). Other resources included non-peer reviewed

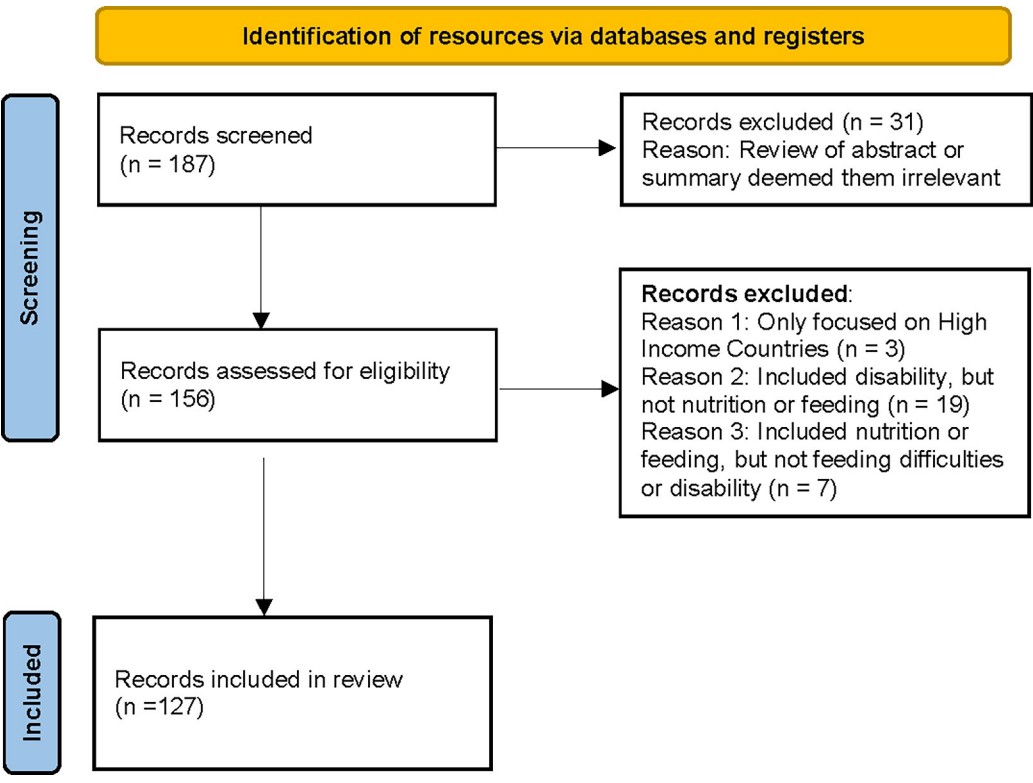

**Fig 1. Eligible resource flowchart.**

reports or program description documents (n = 11, 8.7 percent) and program resources, such as training materials (n = 20, 15.7 percent). Twenty-two percent (n = 28) of resources were related to programs or specific interventions.

## Key informant interviews

A purposive sampling approach was used for key informant interviews (KIIs). We developed a list of diverse experts in the areas of nutrition, feeding difficulties, and disability. Additional key informants were identified in the desk review or via snowball sampling. Using a semi-structured interview guide, interviews were conducted remotely and were either individual interviews or interviews with a group from the same organization. Interviews were recorded and the study team took notes. Names have been withheld to protect privacy.

**Participant characteristics.** The research team invited 57 key informants to be interviewed and completed interviews with 42 (74 percent response rate). Descriptive statistics of the key informants are summarized in Table 1.

## Analysis

Reading and interview notes were organized using two structured matrices based on challenges and opportunities in these areas: (1) the prevalence of feeding difficulties; (2) identification of children with feeding difficulties; and (3) support to manage feeding difficulties. The universal progressive model of care framework for services outlined in the Nurturing Care Framework [15] was used to organize our findings from universal services for all children up to indicated services for children requiring specialized care with referral processes to move

**Table 1. Descriptive characteristics of key informants.**

| Descriptive Characteristic | n | % |
|---|---|---|
| Total | 42 | 100.0 |
| Sex | | |
| Female | 36 | 86.0 |
| Male | 6 | 14.0 |
| Place of Work | | |
| Government | 7 | 17.0 |
| Nongovernmental Organization (NGO) | 22 | 52.0 |
| United Nations Agencies | 3 | 7.0 |
| Academia | 10 | 24.0 |
| Professional Background | | |
| Nutrition | 8 | 19.0 |
| Medicine | 13 | 31.0 |
| Public Health | 3 | 7.0 |
| Disability or Rehabilitation | 18 | 43.0 |
| Region | | |
| South Asia | 6 | 14.0 |
| Europe and Central Asia | 11 | 26.0 |
| Middle East and North Africa | 2 | 5.0 |
| East Asia and the Pacific | 2 | 5.0 |
| North America | 7 | 17.0 |
| Sub-Saharan Africa | 12 | 31.0 |
| Latin America and the Caribbean | 1 | 2.0 |

across levels, which aligns with a twin-track approach for ensuring equity for persons with disabilities (Fig 2). Alongside the universal progressive model of care for service provision, we specifically considered the enabling environment in communities and families for accessing and benefiting from services. Detailed notes from the document review were organized in one

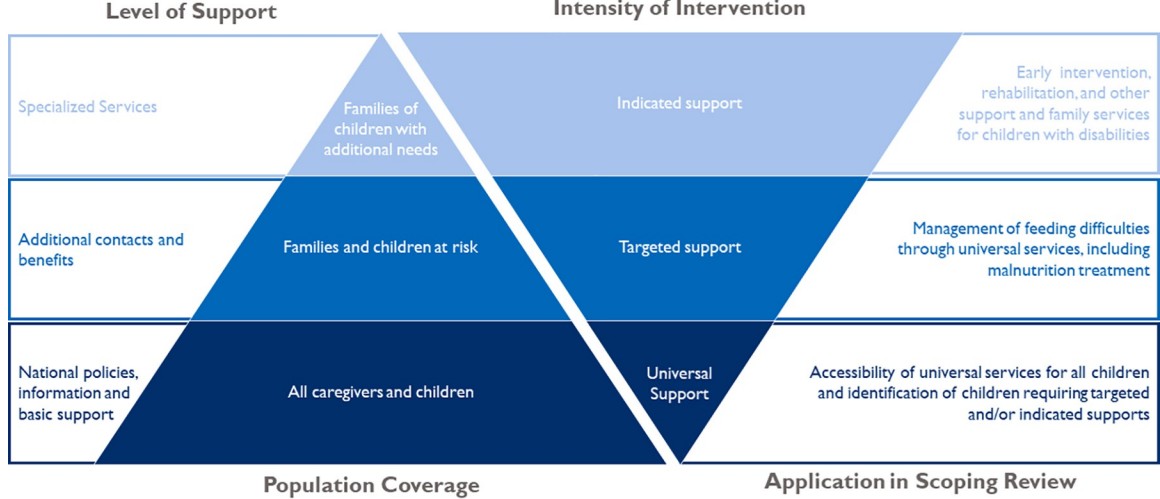

**Fig 2. Application of the universal progressive model.**

matrix, and key themes that emerged from the KIIs were organized into a second matrix. We pulled out the main themes from each matrix, looked for areas of commonality, and used the universal progressive model of care as a framework to organize our findings from the two matrices.

### Ethics

The study was submitted to John Snow, Inc. (JSI) Institutional Review Board and determined to be exempt from review. All key informants provided verbal consent for participation in the study.

## Results

One important finding in both the document review and interviews is the limited research on children with feeding difficulties, and even more limited research on children with disabilities (whether or not they have feeding difficulties). A summary of disability and nutrition programming evidence found limited evidence on the nutritional needs of persons with disabilities and scarce high-quality, rigorous research globally on policy and programming in this area [16]. Key informants for this scoping added that much of the existing research has small sample sizes, research questions that do not include nutritional status, and short study durations. They also said that nutrition studies often exclude children with disabilities, or include them but do not disaggregate the data to understand how they are impacted (Interview with a researcher; Interview with a researcher).

### Identification of feeding difficulties and inclusion in universal services

Identifying a feeding difficulty is the first step in providing adequate care. A number of tools for identifying feeding difficulties in children zero–five years are available, either aimed at health workers or caregivers (Table 2). The tools in Table 2 either address the identification of feeding difficulties alone or the bigger nexus of identifying feeding difficulties among children with disabilities (sometimes within more indicated support services for children with disabilities and their families and sometimes not). For example, the Eating and Drinking Ability Classification System (EDACS) was identified as one valid and reliable system for use in clinical and research contexts for children with cerebral palsy [17].

Although resources exist, our review found that they are not standardized or universally used. Furthermore health workers often lack training to identify and support feeding difficulties in all children, including children with disabilities [16], and funding for capacity strengthening in this area is limited (Interview with a dietitian; Interview with a health worker; Interview with a researcher; Interview with a pediatrician). A review of child development assessment tools found variations in what domains of development are assessed; how tools are adapted, tested, and validated across contexts; and the need for valid tools to identify at-risk children in LMICs [40]. Additionally, many existing tools are not designed to identify disability or provide linked support for families. A report on neurodisability in Africa found an "obvious need to make available simple, practical neurodevelopmental monitoring or surveillance tools that can be integrated with nutritional assessments to benefit children with neurodevelopmental delays or disabilities and nutritional disorders" [41].

Respondents said that health workers accept malnutrition as the norm for children with disabilities or miss malnutrition or feeding issues because they are "hidden" or attributed to the disability (Interview with a pediatrician; Interview with a medical student; Interview with an NGO staff member). One added, "It is a self-fulfilling prophecy: health workers don't treat the child with a disability because they say the child is just going to die. And of course they die

**Table 2. Tools for identifying and managing feeding difficulties.**

| Name of Tool | Authors | Purpose of Tool | Tool Coverage | | Supporting Evidence |
|---|---|---|---|---|---|
| | | | Identification | Management | |
| A Practical Approach to Classifying and Managing Feeding Difficulties [18] | Publication in *Pediatrics Digest* | Classifies feeding difficulties into 3 categories (limited appetite, selectivity, fear of feeding) to determine whether caregivers' concerns are behavioral issues, organic conditions, or based on the caregiver's feeding styles. Meant for health professionals. | X | | No published evaluations. |
| Caring for Children with Developmental Disabilities: A Guide for Parents [19] | Multi-Agency International Training and Support (MAITS) | Provides guidance to support health, well-being, and development of children with disabilities, including symptoms that lead to feeding difficulties, such as difficulties latching, low/high muscle tone, and others. It includes recommendations for health workers who are supporting children with disabilities and an accompanying health worker training program [18]. | X | X | No published evaluations. |
| Holt International's Feeding and Positioning Manual [20] | Holt International | Provides caregivers information about safe feeding practices. Created for children living in underserved communities and orphanage care. It covers feeding fundamentals, feeding across the ages, and feeding development across special populations. Includes a training package for professionals and care providers in institutionalized care settings. | | X | Retrospective analysis of programmatic data of 3,335 children ages 0–18 in Holt International's Child Nutrition Program for children in institutional care in six countries found that 33% of children with disabilities and 54% of children without disabilities who had a feeding difficulty at baseline no longer had a difficulty one year later [21]. |
| SPOON's Feeding and Nutrition Training Package [22] | The SPOON Foundation | A free online training course for caregivers and service providers who work with children with disabilities or children in alternative care. It breaks down the components of mealtime: how to differentiate food textures and how to identify feeding difficulties and opportunities to support caregivers. There is also an app—Count Me In—that provides nutrition and feeding assessments, individualized care plans, and reports to help assess children's growth and feeding and institute changes to their care. | X | X | Preliminary routine programmatic data on 224 children with disabilities in residential care institutions or community-based rehabilitation centers show trends toward improved nutritional status (44% reduction in anemia and 25% reduction in wasting [23]. |
| The Eating and Drinking Ability Classification System (EDACS) [17] | Sussex Community National Health Service Foundation Trust | Classifies eating and drinking performance of people over three years old with cerebral palsy into five levels of ability. Looks at safety and efficiency, with a goal of categorizing conditions for therapy and referring to specialists. | X | | The EDACS provides a valid and reliable system for use in both clinical and research contexts [17], and the Mini-EDACS is being developed for younger children (18–36 months) [24]. |
| Working with Infants with Feeding Difficulties (WIFD) [25] | Multi-Agency International Training and Support (MAITS) | Two-day training for health staff of neonatal intensive care and special-care units in low-resource settings. Fills a gap in training content on feeding issues among premature infants and infants with birth asphyxia and congenital anomalies. | X | X | A quasi- experimental study in neonatal intensive care units in Rwanda found that this package significantly increased early breastfeeding, and exclusive breastfeeding at discharge [26, 27]. |

(*Continued*)

**Table 2.** (Continued)

| Name of Tool | Authors | Purpose of Tool | Tool Coverage | | Supporting Evidence |
|---|---|---|---|---|---|
| | | | Identification | Management | |
| Working with Children with Eating and Drinking Difficulties [28] | Multi-Agency International Training and Support (MAITS) | Eight-day training for health specialists in low resource settings. It includes content related to assessment, goal setting, and interventions for a wide range of eating and drinking difficulties. | X | X | An observational pre-post study in Bangladesh showed improvements in respiratory health, mealtime cooperation, child mood, and caregiver stress. However, there were not significant improvements in growth [29]. |
| Ubuntu (formerly Getting to Know Cerebral Palsy) [30], Juntos [31], and Baby Ubuntu [32] | International Center for Evidence in Disability at the London School of Hygiene and Tropical Medicine | Manuals guide: A participatory, support group approach for caregivers of children with disabilities, including an early intervention package (Baby Ubuntu) for children under age three years. One module focuses on eating and drinking and provides guidance on how to support children when feeding: handling, caregiver frustration, hygiene, dietary needs, positioning, modifying utensils, food textures, and more. The support groups are intended to be facilitated jointly by health workers or rehabilitation professionals including with a parent of a child with a disability. | | X | Pre-post studies of the Ubuntu tools in Ghana, Uganda, and Brazil have shown strong feasibility and acceptability as well as improved feeding practices and positioning, improved quality of life, reduced stigma, and other social benefits, but not improved growth [33–37]. They also show the feasibility of facilitation by a professional and a caregiver [38]. A randomized controlled trial evaluation of Baby Ubuntu is forthcoming [39]. |

because their malnutrition went unaddressed." (Interview with a disability researcher). The combination of challenges in identifying feeding difficulties, limited understanding of the link between disabilities and feeding, and weak or nonexistent referral pathways or specialized services—especially outside major urban areas—puts children with feeding difficulties at a heightened risk of malnutrition (Interview with a speech and language therapist [SLT]; Interview with a physiotherapist; Interview with NGO staff).

Children with disabilities are often not included in universal nutrition services and programs even in countries with large nutrition or early childhood development (ECD) programs, especially in LMICs, due to lack of support for service providers on the needs of children with disabilities and stigma ([40]; Interview with pediatricians; Interview with a researcher). One review looked at 100 clinical trials of ECD interventions, and found that 50 percent of the trials excluded children with disabilities [40]. Food security and nutrition programs are often not designed in accessible ways and do not reach children with disabilities because of cultural stigma, physical barriers, and devaluation of their lives [42, 43]. This is further complicated due to donors prioritizing projects that anticipate seeing growth improvements in children, which cannot be predicted easily or is not the right metric for children with disabilities (Interview with a SLT). Additionally, a number of our respondents mentioned that even when children with disabilities are included in programming, data collection is usually not disaggregated to look at how outcomes might differ for them. A review of 71 national and international guidelines on malnutrition found that while most mention disability, only three had specific sections providing guidance on disability [44].

This review revealed challenges related to knowledge, attitudes, and practices of health workers toward children with disabilities, in general, and to addressing feeding difficulties more specifically. Health workers are expected to provide counseling to correct uncomplicated breastfeeding and complementary feeding difficulties but are not trained to identify when these children or their caregivers need additional support (Interview with a SLT) [26, 27, 45].

There is also limited knowledge on feeding practices for small and sick newborns, who are at greater risk for feeding difficulties and malnutrition [46]; limited skills and time for health workers to provide counseling that allows for "trial and error" of different interventions [47]; and limited use of existing resources, such as guidelines for managing challenges related to cleft lip/palate [48]. In addition, primary health care is often overstretched and underfunded, which affects the quality of support services for all children. One respondent shared, "You'll be lucky to get quality care for an acute health problem. If you're lucky you get one counseling session. But these contacts are insufficient." (Interview with pediatrician). In one survey of nutrition programs that addressed children and youth with disabilities in LMICs, common challenges include a lack of trained professionals, lack of funding for disability inclusion, cultural stigma about children with disabilities, and difficulty identifying children with disabilities [42].

## Targeted services for managing feeding difficulties

After feeding difficulties are identified, caregivers need targeted services to manage them, requiring a well-trained workforce with the resources and tools to provide services and support. According to the Academy of Nutrition and Dietetics, there are no specific evidence-based practice protocols to guide nutritional care for intellectual and developmental disabilities and children and youth with special health care needs [49]. This review found tools that support health workers in providing services to address feeding difficulties (see Table 2). Some of these tools have been evaluated and have shown positive results associated with their use, though often only in small-scale, observational studies. For example, observational studies have shown a decrease in feeding difficulties [21], decreased anemia and wasting [23], increased early breastfeeding and exclusive breastfeeding in neonatal care units [27], as well as family benefits such as improvements in mealtime cooperation and caregiver stress [29]. While these are positive outcomes, a number of these evaluations did not find improvements in child growth. Additionally, not all tools have yet been evaluated, and additional evaluations would provide useful information about how to adapt, scale, and use these and other tools in various contexts.

If a child has a known disability, health workers often attribute that child's poor feeding and growth to the disability and do not provide the support that caregivers and children need to manage feeding difficulties, either because they do not know how or do not have sufficient resources [50] (Interview with a developmental pediatrician). One respondent added that health workers frequently do not recognize or address sensory-related feeding problems, such as selective eating or oral hypersensitivity (Interview with a SLT). A review found that better evidence is needed on identifying and managing children with disabilities in malnutrition programs [44], which was supported by respondents (see Box 1). Another report found that "the nutrition sector needs to provide program staff with better guidance and training on how to manage children with neurodisabilities. Further research is needed to understand the needs of nutrition sector staff and develop appropriate training" [34].

### Box 1. Wasting prevention and treatment in children with feeding difficulties

Feeding difficulties can increase children's risk of wasting (low weight-for-height), and children with disabilities and wasting have a higher risk of death than wasted children without a disability. Despite this, a review of guidelines for wasting treatment revealed that few global or country-specific health systems provide guidance on treating wasting in children with disabilities [44].

One qualitative study in Malawi found that feeding difficulties were the biggest issue for health workers who manage severe wasting among children with disabilities. They mentioned intake issues due to poor chewing and swallowing, a lack of available foods of appropriate texture, and food availability in general [51]. The health workers did not receive specialized training on treating children with feeding difficulties and disabilities and severe wasting. In this study, and more generally, our scoping review found that health workers often lack knowledge and skills to manage wasting among children with disabilities. The preferred treatment for outpatient care, ready-to-use therapeutic food, is often too thick and sticky for children with feeding difficulties to consume (Interview with a pediatrician). Alternatives, such as referral to inpatient care for treatment with therapeutic milks, may not be accessible, may require caregivers to spend extensive time away from their families, or may be unsustainable as long-term solutions. They added that if a child's wasting is successfully treated, but the feeding difficulty is not addressed, the child risks relapsing after being discharged from treatment (Interview with a developmental pediatrician). Health workers who work in wasting treatment facilities also said they learned on the job how to support caregivers to improve child feeding (Interview with a pediatrician).

The limited data available from one study found that only 18 percent of children with disabilities treated for wasting were alive at seven-year follow-up, representing seven times greater odds of dying compared to children without disabilities who were treated for wasting [52]. While it is unclear if the children with disabilities were more likely to die because of nutrition and feeding difficulties, this merits further research [52].

## Indicated services for nutritional care of children with disabilities

Availability and access to services that are specifically related to feeding difficulties and often associated disabilities are limited. For example, one respondent mentioned a lack of social services or safety net programs to support families (Interview with a pediatrician). Rehabilitation services are either not available or are of poor quality (Interview with a physiotherapist); are often disconnected from the health system (Interview with a SLT); and do not have a specialized workforce to support children with disabilities, including speech and language therapists and occupational therapists who have experience in managing feeding difficulties (Interview with medical doctors; Interview with a SLT; Interview with NGO staff). Where there is a workforce, they often work in the private sector because there is no funding to employ them in the public sector (Interviews with SLTs), or they may work in major urban areas but not reach districts or communities farther away (Interview with a physiotherapist; Interview with a psychologist). Travel costs often make these services prohibitive, and referral systems are weak (Interview with a physiotherapist; Interview with a physiotherapist). One study in Ghana found a need for closer links between nutrition and services for persons with disabilities because there are too few referrals from services for persons with disabilities to feeding programs, or vice versa [33]. More broadly, health services in many countries are fragile, poorly coordinated and overstrained, leaving concerns about whether health care workers are available and capable of delivering these services [40].

Support groups for caregivers of children with disabilities are another resource in various settings for providing indicated services. For example, Ubuntu Hub houses three caregiver support group packages, which are listed in Table 2 [53]. These packages have been adapted for use in countries including South Africa, Bangladesh, Ghana, Brazil, and others [30–33]. An

evaluation in Ghana found that in addition to the social benefits of these groups, caregivers who participated reflected on positive changes related to positioning of their children, better feeding practices, and better communication with their children, but they did not see improved growth [33, 34]. They add that the World Health Organization (WHO) recognizes the importance of support networks for the care of children with disabilities, but few studies have evaluated different types of networks and how and why they work to improve caregiver and child outcomes, especially in LMICs. The authors suggest strengthening the model by using caregivers as facilitators or co-facilitators in the training and note that this approach was taking place in Uganda for children with neurodevelopmental conditions, and in Brazil with children with Zika [33]. A small study of this methodology in Brazil found that using paid expert mothers as facilitators was a feasible approach for peer learning and was acceptable to participants and facilitator therapists. They were seen as important to group sharing and learning and provided support and encouragement [38].

Several respondents emphasized the need for government commitment to provide, scale up, and sustain disability services, which are instead often provided by civil society or donor organizations (Interview with a physiotherapist; Interview with a SLT; Interview with a disability expert). Examples of a promising practice for strengthening government systems includes building a pipeline of speech and language therapists through a local university training program in Ghana, and working with government to overcome barriers to integration of this new workforce into routine services (Interview with a SLT); integration of follow-up care for small and sick newborns including specialized feeding and developmental support in Rwanda [47] (Interview with a nurse); and working with community health workers to help facilitate identification of children with disabilities and provide initial support services (Interviews with a pediatrician, occupational therapist, physiotherapist, and psychologist).

## Enabling environment in communities and families

Families of children with feeding difficulties face challenges providing the care their children need, including coping with high-care demands, accessing guidance and support from services at all levels, obtaining appropriate foods, and combating stigma. Caregivers of children with feeding difficulties need guidance on what food to give their children and how to prepare it (e.g., smaller snacks more frequently, rather than three meals per day or modified textures) and support to provide responsive feeding (Interviews with a SLT, nurse, and researcher).

Difficulties swallowing and chewing or sensory challenges that make certain textures hard to eat may inhibit children with feeding difficulties from eating typical table foods. In many settings, food that is recommended for children with feeding difficulties is unavailable or, if available, is more expensive or requires adaptations such as modifying textures or utensils (Interview with a SLT). One health worker in Zimbabwe stressed the need to prioritize locally available, nutritious foods, and to teach caregivers how to prepare them for children with feeding difficulties (Interview with a pediatrician). Numerous respondents mentioned a lack of assistive products, such as supportive seating to improve posture or blenders to modify food and drink textures (Interviews with a pediatrician, a registered dietitian, and director of an Organization of Persons with Disabilities [OPD]). Even when there are ideas for local solutions, such as adaptive equipment, there is not enough funding to develop and test them (Interview with a registered dietitian) or to scale them up (Interview with a researcher).

Given societal barriers and discrimination on the basis of disability, having a child with one or more feeding difficulties that also has a disability causes financial strain on families; caregivers are often unable to earn income as their child's needs are constant and unsupported (Interview with a medical doctor; Interview with a director of an OPD). One study found that

feeding children with cerebral palsy can take caregivers up to seven hours a day, which can cause high stress and contribute to less responsive, or abusive, feeding practices [29]. Support services are often inaccessible due to financial barriers, insufficient supply of services, lack of expertise, lack of time, and competing priorities (Interviews with a nurse, pediatrician, and NGO staff). One Ugandan solution mentioned in interviews was day care rehabilitation centers run by caregivers of children with disabilities. The centers provided a safe place for children with disabilities to receive care and supportive services while caregivers worked. They also provide an alternative to institutionalization (Interview with a director of an OPD).

Stigma and shame associated with disability contributes to isolation of the child and family and may delay diagnosis and access to feeding and other services (Interview with an occupational therapist). Caregivers often don't bring their child with a disability to participate in community events for fear of bringing shame upon the family [54]. One respondent said that children with disabilities are often "hidden" by their families and that data and funding for these children are needed (Interview with a registered dietitian). One study in Peru found that caregivers were not provided emotional support to prepare for having a child with a disability, and abandonment by the father was common [55]. Caregivers said that disability is considered a disease to be treated, rather than a social condition in which children and their families have rights. Providers in the study perceived children with disabilities as neglected by their caregivers and commonly held the mother responsible for delays. Some health care providers showed indifference toward children with disabilities and little willingness to care for them [56]. Some communities practice infanticide of infants with visible disabilities by discouraging caregivers from feeding them [56] (Interview with NGO staff). Another respondent said that children with disabilities are often the last to be prioritized or given food in food insecure households (Interview with a pediatrician). Stigma and lack of support or services exacerbates the vulnerability of these children to malnutrition and increases the likelihood that their feeding difficulties will go unaddressed.

Multiple respondents stated that if stigma is not addressed, no other services or interventions will adequately support children with disabilities (Interview with an SLT and dietitian). One respondent added that education and counseling services around how to address stigma —coupled with supportive social safety net solutions—are key (Interview with a nutritionist). A Ugandan disability advocate with lived experience of disability, discussing social inclusion, stated, "If this message can be spread, then tomorrow we will not have people and society that stigmatize persons with disabilities. This younger generation, these young trees, will grow knowing that having a disability is normal, that I need to support my friend with a disability. So then inclusion is not an issue. The problem is the grown up trees are hard to change; they do not accept." (Interview with a director of an OPD).

## Discussion

This review highlights the dearth of programs and evidence to identify and support children with feeding difficulties and their families in LMICs across all levels of the universal progressive model of care. While some resources and promising approaches exist, they are not standardized or universally used, staff are not trained to use them, and there is insufficient funding to implement them. The combination of challenges in identifying feeding difficulties, a lack of understanding of the link between disabilities and feeding, and weak or nonexistent referral or specialized services puts these children at risk of malnutrition. Additionally, their families face challenges providing the care they need, including coping with high care demands, accessing support, obtaining appropriate foods, and managing stigma. Based on the findings, we have four areas of recommendations for strengthening the identification of feeding difficulties and service provision to meet the needs of children with feeding difficulties—whether or not they

are disabled—and their caregivers. These recommendations are to: (1) build the evidence base, (2) strengthen systems to improve identification and service provision; (3) provide direct support to families; and (4) conduct advocacy.

The first recommendation is to *build the evidence base* on interventions for identifying and supporting children with feeding difficulties and their families, and evaluate the existing tools and approaches to better understand which ones have or could have the greatest impact on feeding difficulties, especially in LMICs. One finding was that feeding-related interventions for children with disabilities have not seen improvements in child growth and nutritional status, despite helping children with chewing and swallowing through postural support, decreasing stress for caregivers, and making mealtimes more enjoyable. The literature and key informants recommended doing more and better research, especially in LMICs, to find what works to support caregivers to care for children with feeding difficulties. Specific efforts to build the evidence base should include conducting formative research with caregivers of children with feeding difficulties to prioritize approaches that directly respond to their needs, conducting implementation research ideally leveraging existing tools highlighted in this review as a starting point, and intentionally including and tracking children with feeding difficulties and disabilities in implementation research and routine data systems. The Washington Group Question Sets provide a simple survey tool for identifying children ages two years and up with a disability [57], and the recently launched UNICEF Centre of Excellence on Data for Children with Disabilities is a promising approach to better data [58]. Additionally, there needs to be more evidence generated on the efficacy and effectiveness of tools for the identification and management of feeding difficulties and interventions that overcome the barriers to the uptake of these tools. Finally, research exploring what interventions impact growth and nutritional status for children with disabilities is also necessary, as those outcomes have been the hardest to achieve. This overarching need for evidence has implications for the other review recommendations, which include solutions that have not all been tested in relation to children with feeding difficulties or disabilities but were seen during the literature review or were recommended by respondents.

Next, *strengthened health systems* are needed to improve identification and service provision for children with feeding difficulties in health services at all levels of the universal progressive model of care. This scoping review revealed gaps across all six WHO health systems building blocks for children with feeding difficulties and disabilities (Table 3) [59]. Promising practices identified in the review were primarily focused on strengthening service delivery, such as capacity strengthening and workforce development. However, this study focused primarily on service delivery and human resources and additional research into the other building blocks is warranted.

Closing these gaps will require systems strengthening and quality improvement at all levels of service: identifying children who require support; providing support to manage feeding difficulties; and—more holistically—including and supporting children with disabilities in nutrition services, programs, and policies to help them thrive. Specific interventions should include capacity strengthening and addressing misperceptions and biases among health workers and revising guidelines and care protocols related to nutrition and health to include appropriate guidance and support for children with feeding difficulties. Given high rates of wasting and early mortality due to malnutrition among children with disabilities, an opportunity exists to provide guidance on children with disabilities in the forthcoming revision of wasting guidelines [44]. However, the Global Action Plan on Wasting contains only very brief mention of children with disabilities [60]. The current child health redesign process, undertaken by WHO and UNICEF to focus on a life cycle, health promotive approach, offers an ideal opportunity to ensure child health protocols, such as the Integrated Management of Childhood Illness, are revised to better meet the needs of children with disabilities [61].

**Table 3. Gaps in health systems building blocks.**

| | |
|---|---|
| Governance | • NGOs/civil society organizations fund and provide services for persons with disabilities instead of governments<br>• Lack of inclusive policies and programs |
| Information systems | • Disability-disaggregated data not available in nutrition and health services |
| Financing | • Lack of funding for disability-specific programming and disability-inclusion in routine services |
| Service delivery | • Children with disabilities, many of whom have feeding difficulties, may not be included in routine nutrition services<br>• Missed opportunities for early identification and intervention for feeding and nutrition and lack of follow-up structures<br>• Poor quality of rehabilitation services (where they exist)<br>• Lack of guidelines and tools to address feeding difficulties and malnutrition among children with disabilities |
| Medicines and technology | • Assistive products to support feeding unavailable in the health system |
| Workforce | • Limited skills and in-service training opportunities related to feeding difficulties or supporting children with disabilities among primary health care workers<br>• Lack of specialized workforce and job pipelines for rehabilitation trainees<br>• Stigma and attitudinal barriers among health providers |

To address social determinants and family supports that affect nutrition outcomes for children with feeding difficulties and disabilities, *families need direct support*. Currently, support for caregivers is limited, and what does exist is not available at scale. Poverty and high stress among families is compounded by the lack of available, accessible, appropriate, and affordable services that are inclusive and of high quality. Experience to draw from is limited. Formative research and collaboration with families can help identify ways to strengthen social support to families and increase their access to services. Opportunities include establishing peer-to-peer support groups, providing access to assistive products using local resources developed by trained personnel, and providing standardized guidance on caring for children with disabilities and feeding difficulties in food distribution programs. Policymakers could ensure children with feeding difficulties and disabilities and their families are priority groups for eligibility criteria in social protection schemes and food supplementation programs. As services to support children with disabilities and their families in LMICs are expanded, such as the WHO Caregivers Skills Training [62], it will be essential to integrate support around feeding and nutrition.

Finally, *advocacy* is needed to raise awareness of the need and opportunities to support children with feeding difficulties and disabilities. This includes addressing a lack of support and high levels of stigma at the community level, advocating for inclusive and sufficient services at the health systems level, advocating for national-level policies, and inclusive global agendas and strategies. Advocacy is essential along with strengthening systems and providing direct support to families and individuals and can happen simultaneously. Throughout the scoping review, documents and respondents reiterated that without advocacy, systems strengthening and direct family support will be insufficient to address feeding difficulties. This is particularly relevant for children with disabilities who face the most stigma and exclusion. However, it is also needed to ensure that disability and skilled support for feeding difficulties are part of global and national nutrition agendas. At the policy level, we must ensure disability-inclusive nutrition programs are included in government- and donor-funded guidance documents, program descriptions, or other nutrition-related resources. At the community level, we should promote uptake of Community-Based Inclusive Development (CBID) approaches to provide information on the causes, needs, and care of children with disabilities.

## Limitations

There were limitations when preparing this manuscript. First, the group conducting this review did not include individuals who identify as having a disability or individuals from LMICs. We looked for interview participants from diverse regions and countries; however, we were only able to interview one respondent from the Latin America and the Caribbean region, and a large number of respondents were from high-income countries in Europe and North America. We did not directly interview caregivers of children with disabilities. We did not do a comprehensive review of national or international nutrition policies, nor did we look directly at how systems other than health systems impact children. Our review included children with feeding difficulties who may or may not have a disability; it was at times challenging to differentiate these groups of children given the high rates of feeding difficulties among children with disabilities. However, most recommendations would benefit all these children by making services more capable of responding to feeding difficulties and more inclusive of all children. In addition, this was not a systematic review given the goals to more broadly assess the current landscape in terms of both peer-reviewed literature and non-peer reviewed materials; a systematic review of standard databases for peer reviewed literature that includes an assessment of the quality of each study included may be more appropriate in the coming years as hopefully more peer-reviewed evaluations of early intervention and disability-inclusive nutrition programs become available. Finally, this is a global review, so we are limited in our ability to generalize perspectives of our key informants.

## Conclusion

This scoping review found insufficient support for children with feeding difficulties and their families. While some resources and promising system strengthening approaches to identify and support these children exist, they are not standardized or universally used, and not enough funding is available to implement them. The combination of challenges in identifying children with feeding difficulties and disabilities and providing targeted and indicated services for them puts them at risk of malnutrition. Additionally, their families face challenges accessing support and providing the care they need, including coping with high care demands, obtaining appropriate foods, and overcoming stigma for children with disabilities. Four areas of recommendations emerged from this work: (1) Build the evidence base for effective interventions to identify and support these children and their families; (2) Strengthen health systems to improve identification and service provision, and expand the availability of feeding services; (3) Provide direct support to families to address determinants that affect nutrition outcomes; and (4) Conduct advocacy to raise awareness of the needs and opportunities related to children with feeding difficulties and disabilities.

## Supporting information

**S1 Checklist. Preferred Reporting Items for Systematic reviews and Meta-Analyses extension for Scoping Reviews (PRISMA-ScR) checklist.**
(PDF)

## Acknowledgments

We are very grateful to the key informants who shared their time, experience, and expertise to inform this scoping review. We would also like to thank colleagues who provided guidance throughout this scoping review, including Jamie Gow, Katherine Guernsey, Kirsten Lentz,

Joshua Josa, Kathryn Beck, Jennifer Yourkavitch, Altrena Mukuria-Ashe, Lori Baxter, and Christine Kirungi.

**Disclaimer:** This review uses "persons with disabilities" throughout, consistent with the Convention on the Rights of Persons with Disabilities and the USAID Disability Communication TIPS. The authors acknowledge that many members of the disability community prefer identify-first language (e.g., "disabled person").

## Author Contributions

**Conceptualization:** Alyssa Klein, Malia Uyehara, Madina Olomi, Catherine M. Kirk.

**Data curation:** Alyssa Klein, Malia Uyehara, Andrew Cunningham, Madina Olomi, Catherine M. Kirk.

**Formal analysis:** Alyssa Klein, Malia Uyehara, Andrew Cunningham, Madina Olomi, Catherine M. Kirk.

**Investigation:** Alyssa Klein.

**Methodology:** Catherine M. Kirk.

**Project administration:** Alyssa Klein, Malia Uyehara.

**Resources:** Alyssa Klein, Andrew Cunningham, Madina Olomi, Catherine M. Kirk.

**Supervision:** Kristen Cashin, Catherine M. Kirk.

**Visualization:** Alyssa Klein, Catherine M. Kirk.

**Writing – original draft:** Alyssa Klein, Kristen Cashin, Catherine M. Kirk.

**Writing – review & editing:** Alyssa Klein, Malia Uyehara, Andrew Cunningham, Madina Olomi, Kristen Cashin.

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
