## [Decision Letter · Decision Letter 0]

12 Dec 2022

PGPH-D-22-01381

Nutritional Care for Children with Feeding Difficulties and Disabilities: A Scoping Review

Dear Dr. Klein,

Thank you for submitting your manuscript to PLOS Global Public Health. After careful consideration, we feel that it has merit but does not fully meet PLOS Global Public Health’s publication criteria as it currently stands. Therefore, we invite you to submit a revised version of the manuscript that addresses the points raised during the review process.

We look forward to receiving your revised manuscript.

Kind regards,

Jitendra Kumar Singh, PhD

Academic Editor

Journal Requirements:

1. Please send a completed 'Competing Interests' statement, including any COIs declared by your co-authors. If you have no competing interests to declare, please state "The authors have declared that no competing interests exist". Otherwise please declare all competing interests beginning with the statement "I have read the journal's policy and the authors of this manuscript have the following competing interests:"

2. Please provide separate figure files in .tif or .eps format.

3. We noticed that you used “data not shown”/"unpublished data" in the manuscript. We do not allow these references, as the PLOS data access policy requires that all data be either published with the manuscript or made available in a publicly accessible database. Please amend the supplementary material to include the referenced data or remove the references.

4. We have noticed that you have uploaded Supporting Information files, but you have not included a list of legends. Please add a full list of legends for your Supporting Information files after the references list. 

5. In the online submission form, you indicated that your data will be submitted to a repository upon acceptance.  We strongly recommend all authors deposit their data before acceptance, as the process can be lengthy and hold up publication timelines. Please note that, though access restrictions are acceptable now, your entire data will need to be made freely accessible if your manuscript is accepted for publication. This policy applies to all data except where public deposition would breach compliance with the protocol approved by your research ethics board. If you are unable to adhere to our open data policy, please kindly revise your statement to explain your reasoning and we will seek the editor's input on an exemption. Please be assured that, once you have provided your new statement, the assessment of your exemption will not hold up the peer review process.

Additional Editor Comments (if provided):

The first paragraph of the discussion should contain a summary of the study results and the main findings and then recommendations.

Reviewers' comments:

Reviewer's Responses to Questions

**Comments to the Author**

1. Does this manuscript meet PLOS Global Public Health’s publication criteria? Is the manuscript technically sound, and do the data support the conclusions? The manuscript must describe methodologically and ethically rigorous research with conclusions that are appropriately drawn based on the data presented.

Reviewer #1: Yes

Reviewer #2: Partly

2. Has the statistical analysis been performed appropriately and rigorously?

Reviewer #1: N/A

Reviewer #2: N/A

3. Have the authors made all data underlying the findings in their manuscript fully available (please refer to the Data Availability Statement at the start of the manuscript PDF file)?

Reviewer #1: Yes

Reviewer #2: Yes

4. Is the manuscript presented in an intelligible fashion and written in standard English?

Reviewer #1: Yes

Reviewer #2: Yes

5. Review Comments to the Author

Reviewer #1: Thank you for giving me the opportunity to review this paper. I learned a lot from reading it; it is an important but weakly investigated issue that this paper gives very welcome attention to. Many congratulations to the authors.

There are two main areas where the paper needs strengthening:

1) It is inconsistent as to whether it focuses only on feeding difficulties in disabled children, or also covers feeding difficulties in non-disabled children

2) The findings need to be presented in a clearer framework. This will make the findings and discussion sections sharper.

Detailed comments:

Abstract

- Methods: you state that “findings were organized using structured matrices of challenges and opportunities in the identification, management, and support for children with feeding difficulties, disabilities, and their families”. This is not quite how you structure the findings, however (see later comments)

Introduction

- You frame this as about disability (first few paragraphs) but then you say (line 87) that you cover non-disability related feeding difficulties; reinforced by line 91 (especially children with disabilities). I’m not sure this is consistent (see later comments). If it is right, you should include a section on non-disability related feeding difficulties – what they are – and reconsider whether the initial overarching framing should be about disability or about feeding difficulties with disability as a secondary theme.

- Is the figure in line 76/77 just for LMICs, or global? Clarify.

- Line 85-88, I would clarify that you have only focused this review on LMICs.

Methods

- What is a non-systematic review – can you define and it provide a reference?

- Line 142: you have a slightly different framing of how you organize the results than appears in the abstract (prevalence, identification, support to manage).

- If you have separate matrices for the document review and the KIIs, can you explain how you synthesized across the two?

Results – Figure 1

- I think lines 153-162 are really about methods and should come in your methods section on the desk review (i.e. around line 129)

- Explain the rationale and process for dropping the first 31

- You say here (reason 3) that you exclude studies that talk about nutrition/feeding but not disability. This implies that you only include studies that mention disability. But this is inconsistent with the framing above. And also inconsistent with your framing in line 163/4.

- I think it would be helpful to say how many studies/documents were about challenges facing children, and how many are about actual interventions. This would help you be clearer on the division with promising practices, later on

Results – Document

- You need to be clear (line 163) on whether this finding is a conclusion you draw from the search (we didn’t find much robust research) or whether this is a finding that is contained in the documents you reviewed/was mentioned by the KIIs. If it is the former, you will need to define ‘robust’ and ‘quality’ in this context and how you made the judgement.

Results – Participant Characteristics

- I would shift lines 175-180 to the methods section about KIIs

Results – Health Systems

- To be honest, I struggled a little with the structure by which you’ve presented the results.

- Table 2 is like a summary table of how the findings relate to the WHO building blocks – but you haven’t introduced the concept of the building blocks (or said up front you will analyze the data this way). You don’t use this framing in the subsequent text. I did wonder if you would be better off moving this to the discussion as a way of pulling together some of the findings when you make your recommendation 2. If you moved it there, this would add value. At the moment it confuses the narrative.

- You then structure by identification and then management, and then have a completely new section for communities/family. This is slightly different again from all of your previous framings.

Results – Identification of feeding difficulties

- Table 3 includes some tools that are specific to the disabled and some that are not. Whether this is a problem will depend on the earlier issue of being precise on what you are covering.

- In the text, you should have a para on the evidence base (that is included in the final column on the table) about the evidence on the potential impact of using the tools. Including the limitations of the evidence.

- I’m just noting that all of the content in this section is really about challenges in the tools being used as designed, but you state elsewhere that you also want to include promising practices (which are not mentioned here, except for the evidence on the tools themselves).

- If there are no promising practices aside from the tools in the table (e.g. no evidence on complementary system strengthening interventions to drive uptake of tools) then you need to say so explicitly. This comment applies to the next section also.

Results – Services, tools, and resources for managing feeding difficulties

- A couple of typos (e.g. line 257)

- I don’t think (umbrella comment) adding dates for all of the interviews in brackets is necessary – it doesn’t add anything and undermines readability.

- These are also all about problems, and nothing about interventions to overcome them

Results – Communities and families

- The first three sentences are inconsistent about whether they are talking about generic feeding difficulties, or only for disabled. This is an example where more precision would help (this is an overarching comment).

- Only in this section do you have examples of promising practices

Discussion

- You should sharpen this based on the comments above (e.g., the fact that you have identified no examples of system level interventions to drive the uptake of tools)

- On the first recommendation, you may wish to be clear that there needs to be evidence generated on i) the efficacy and effectiveness of tools for identification and management, and ii) interventions that overcome the barriers to the uptake of these tools

- On the second recommendation, I would bring your table etc. from the start of the results here, it would flow nicely

Limitations

- It would be good to reflect on the limits of doing a non-systematic review

Conclusion

- This is likely to need some tweaking based on the edits above.

Reviewer #2: This is a well-written manuscript addressing programs supporting nutritional care of children with disability and non-disability related feeding difficulties. I congratulate the author(s) to this attempt, as this is an important and understudied topic. Moreover, it is really challenging to conduct a scoping review on this topic, since selecting literature search terms that will thoroughly capture relevant research articles on this topic is not entirely straightforward. Though the authors has prepared the manuscript comprehensively and makes a moderately useful contribution to the subject, it also has scope for improvement in the definition of its scope and goals, in the description of the methods which is incomplete, I feel within this manuscript. Following are some of the concerns for improvement.

(a) The abstract has not followed standard/recommended structure, and scoping review period (Dec 2020-May 2022) and abbreviations, needs to spell out on first use (viz., LMICs, USAID)

(b)Though the objective of the review is clear and the scoping review methodology used is appropriate, the methods section does not provide enough information to know exactly the review work to identify peer-reviewed articles conducted (to identify 187 potential resources) using generic web searches (Google Scholar) without using electronic databases (PubMed, Medline, EMbase, Cochrane Library, Web of Science etc.) and at least a brief summary of the approach is needed.

(c) the authors suggested to revisit the literature search strategy, to explore the existing literature further, and assess whether additional evidence would be pulled with additional targeted search terms, and consider revising the study to re-run the search and add additional evidence to the analysis.

(d) the quality of studies needs to be further assessed since the studies considered are for short period

In conclusion, the subject addressed in this manuscript is worth of investigation and recommend for possible consideration as this is an important contribution to the literature, and the findings provide interesting and important knowledge to the research and practice community. However, before taking a decision for possible consideration of publication, the manuscript needs minor revision on its structure, justification on data sources, results and completeness.

6. PLOS authors have the option to publish the peer review history of their article (what does this mean?). If published, this will include your full peer review and any attached files.

**Do you want your identity to be public for this peer review?** For information about this choice, including consent withdrawal, please see our Privacy Policy.

Reviewer #1: No

Reviewer #2: No

---

## [Editor Report · Decision Letter 1]

15 Feb 2023

Nutritional Care for Children with Feeding Difficulties and Disabilities: A Scoping Review

PGPH-D-22-01381R1

Dear Ms. Klein,

We are pleased to inform you that your manuscript 'Nutritional Care for Children with Feeding Difficulties and Disabilities: A Scoping Review' has been provisionally accepted for publication in PLOS Global Public Health.

Best regards,

Jitendra Kumar Singh, PhD

Academic Editor